# Automatic Evaluation of Attribution by Large Language Models

**Xiang Yue    Boshi Wang    Ziru Chen    Kai Zhang    Yu Su    Huan Sun**

The Ohio State University

{yue.149,wang.13930,chen.8336,zhang.13253,su.809,sun.397}@osu.edu

## Abstract

A recent focus of large language model (LLM) development, as exemplified by generative search engines, is to incorporate external references to generate and support its claims. However, evaluating the attribution, i.e., verifying whether the generated statement is fully supported by the cited reference, remains an open problem. Although human evaluation is common practice, it is costly and time-consuming. In this paper, we investigate automatic evaluation of attribution given by LLMs. We begin by defining different types of attribution errors, and then explore two approaches for automatic evaluation: prompting LLMs and fine-tuning smaller LMs. The fine-tuning data is repurposed from related tasks such as question answering, fact-checking, natural language inference, and summarization. We manually curate a set of test examples covering 12 domains from a generative search engine, New Bing. Our results on this curated test set and simulated examples from existing benchmarks highlight both promising signals and challenges. We hope our problem formulation, testbeds, and findings will help lay the foundation for future studies on this important problem.[1]

## 1 Introduction

Generative large language models (LLMs) (Brown et al., 2020; Ouyang et al., 2022; Chowdhery et al., 2022; OpenAI, 2023a,b, *inter alia*) often struggle with producing factually accurate statements, resulting in hallucinations (Ji et al., 2023). Recent efforts aim to alleviate this issue by augmenting LLMs with external tools (Schick et al., 2023) such as retrievers (Shuster et al., 2021; Borgeaud et al., 2022) and search engines (Nakano et al., 2021; Thoppilan et al., 2022; Shuster et al., 2022).

Incorporating external references for generation inherently implies that the generated statement is backed by these references. However, the validity of such attribution, i.e., *whether the generated statement is fully supported by the cited reference*, remains questionable.[2] According to Liu et al. (2023), only 52% of the statements generated by state-of-the-art generative search engines such as New Bing and PerplexityAI are fully supported by their respective cited references.[3]

Inaccurate attribution compromises the trustworthiness of LLMs, introducing significant safety risks and potential harm. For instance, in healthcare, an LLM might attribute incorrect medical advice to a credible source, potentially leading users to make harmful health decisions. Similarly, in finance, faulty investment advice attributed to a reliable source may cause substantial financial losses.

To identify attribution errors, existing attributed LLMs (Nakano et al., 2021; Thoppilan et al., 2022) rely heavily on human evaluation, which is both expensive and time-consuming. For instance, the average cost of annotating a single (query, answer, reference) example is about $1 in Liu et al. (2023). In the actual use of attributed LLMs, it is the user who needs to be wary of the attribution and manually verify it, which puts a tremendous burden on their side. Therefore, effective and reliable methods to automatically evaluate attribution and identify potential attribution errors are highly desired.

Towards this goal, we take the first step by introducing *AttrScore* (Figure 1), a framework designed for automatic evaluation of attribution and identification of specific types of attribution errors. We propose a new problem formulation that categorizes attribution into three types: 1) *attributable*: the reference fully supports the generated statement; 2) *extrapolatory*: the reference lacks sufficient information to support the generated state-

---

[1]Our code and dataset are available at: https://github.com/OSU-NLP-Group/AttrScore

[2]*Attribution* primarily refers to "the act of attributing something" in this paper, which is similar to "verifiability" as defined in Liu et al. (2023).

[3]www.bing.com/new, www.perplexity.ai

Figure 1: We make the first step towards automatically evaluating attribution and identifying specific types of errors with AttrScore. We explore two approaches in AttrScore: (1) prompting LLMs, and (2) fine-tuning LMs on simulated and repurposed datasets from related tasks.

ment, and 3) *contradictory*: the generated statement directly contradicts the cited reference. Unlike existing work (Bohnet et al., 2022) that uses binary categorization (i.e., attributable or not) and Liu et al. (2023) that defines the degree of reference support for the generated statement as "full", "partial", or "no support", our fine-grained error categorization aids humans in better understanding the type of an attribution error made by an LLM. This not only enhances safe system usage but also provides valuable insights for future development of mechanisms tailored to correct specific errors.

We explore two approaches in AttrScore: 1) prompting LLMs and 2) fine-tuning LMs on simulated and repurposed data from related tasks such as question answering (QA), fact-checking, natural language inference (NLI), and summarization. For evaluation, unlike existing work (Liu et al., 2023; Gao et al., 2023) that only uses queries from existing benchmarks, we curate a set of test examples covering 12 different domains from a generative search engine, New Bing. This is the first evaluation set for measuring the attribution of LLMs with queries created based on real-life interactions,

hence avoiding the data contamination issue.

Our results indicate that both approaches show reasonable performance on our curated and simulated test sets; yet there is still substantial room for further improvement. Major sources of evaluation failures include insensitivity to fine-grained information comparisons, such as overlooking contextual cues in the reference, disregard for numerical values, and failure in performing symbolic operations. In light of these findings, we discuss potential directions for improving AttrScore, including training models to be more strongly conditioned on the reference, and augmenting them with external tools for numerical and logical operations.

With the new formulation of attribution errors, the development of AttrScore, the introduction of new test sets, and the insights into challenges and potential directions for future work, we hope our work can help lay the foundation for the important task of automatically evaluating LLM attributions.

## 2 Problem Formulation

The primary task in this paper is to evaluate attribution, which involves verifying whether a reference

provides sufficient support for a generated answer to a user's query. Our task setting prioritizes one reference per statement, a unit task that more complex scenarios can be decomposed to. We study such a setting as it forms the basis for dealing with multiple references or distinct segments (Liu et al., 2023; Gao et al., 2023).

Prior work, such as Rashkin et al. (2021); Gao et al. (2022); Bohnet et al. (2022), mainly focuses on binary verification, i.e., determining if a reference supports the generated answer or not. We propose advancing this task by introducing a more fine-grained categorization. Specifically, we classify attributions into three distinct categories:[4]

- **Attributable**: The reference fully supports the generated answer.

- **Extrapolatory**: The reference lacks sufficient information to validate the generated answer.

- **Contradictory**: The generated answer contradicts the information presented in the reference.

To illustrate, consider a contradictory example (Figure 1). The query is *"What was the unemployment rate in Germany in 2020?"*, and the generated answer is *"4.31%"*. However, the reference states that the rate was *"3.81%"*, contradicting the generated answer. An extrapolatory instance, on the other hand, would be a query about the *"gas price in California"*. While the reference is relevant, it does not contain specific information to verify the correctness of the generated answer.

Following these examples, we see the importance of granularity in error classification. A fine-grained classification allows us to pinpoint the nature of the errors, be it contradiction or extrapolation. Users can better understand the type of errors an LLM might make, enabling them to use the model more safely. Additionally, such an error identification system can guide future training processes of attributed LLMs, leading to specific mechanisms' development to correct such errors.

Our categorization also offers a departure from the existing approach (Liu et al., 2023), which emphasizes on degree of support ("full", "partial", or "none") rather than attribution error types. Our approach highlights specific issues in attribution

evaluation for more effective error management and system improvement.

Formally, the task of attribution evaluation involves a natural language query $q$, a generated answer $a$, and a reference $x$ from an attributed LLM. The goal is to develop a function, denoted as $f$, that inputs $(q, a, x)$ and outputs a class label indicating whether "according to $x$, the answer $a$ to the query $q$ is attributable, extrapolatory or contradictory."[5]

## 3 Automatic Evaluation of Attribution

Following our problem definition, we introduce two approaches for automatic evaluation of attribution: prompting LLMs and fine-tuning LMs on simulated and repurposed data from related tasks.

### 3.1 Prompting LLMs

Recent research (Fu et al., 2023) has demonstrated the possibility of prompting LLMs to evaluate the quality of generated text using their emergent capabilities (Wei et al., 2022b), such as zero-shot instruction (Wei et al., 2022a) and in-context learning (Brown et al., 2020). Following this approach, we prompt LLMs, such as ChatGPT (OpenAI, 2023a), using a clear instruction that includes definitions of the two types of errors (as shown in Figure 1) and an input triple of the query, answer, and reference for evaluation. The complete prompt used in our study can be found in Appendix Table 6.

### 3.2 Fine-tuning LMs on Repurposed Data

The primary challenge in fine-tuning LMs for automatic attribution evaluation is the lack of training data. One potential approach is to hire annotators to collect real samples, but the cost can be prohibitive.

Here, we first repurpose datasets from three related tasks (fact-checking, NLI, and summarization). We then propose to further simulate more realistic samples from existing QA benchmarks.
**Repurpose data from fact-checking, NLI, and summarization tasks.** Given the connections between our attribution evaluation task and the tasks of fact-checking, NLI, and summarization, we propose to utilize datasets from these fields to enrich our training examples. Fact-checking data and NLI data, with their emphasis on assessing the consistency and logical relationship between claims (hypothesis) and evidence (premise), mirrors our task's

---

[4]We acknowledge that while these categories are generally mutually exclusive, complex scenarios might blur the boundaries between them. However, such cases are very rare. For the purpose of this study, we maintain their exclusivity to enable clear and focused error analysis.

[5]It is important to note that this evaluation focuses on the "verifiability" of the answer based on the reference. It does not measure the "relevance", i.e., whether the answer correctly responds to the query (Liu et al., 2023).

| **Query:** Which apostle had a thorn in his side? | **Query:** Which apostle had a thorn in his side? | **Query:** Which apostle had a thorn in his side? | **Query:** Which apostle had a thorn in his side? |
|---|---|---|---|
| **Short Ans:** Paul [1] | **Short Ans:** Phillip [1] | **Short Ans:** Paul [1] | **Short Ans:** Paul [1] |
| **Long Ans:** Paul was an apostle who had a thorn in his side [1]. | **Long Ans:** Phillip had a thorn in his side [1]. | **Long Ans:** Paul was an apostle who had a thorn in his side [1]. | **Long Ans:** The apostle who had a thorn in his side is Paul [1]. |
| **References** | **References** | **References** | **References** |
| [1] en.wikipedia.org/wiki/ Thorn_in_the_flesh | [1] en.wikipedia.org/wiki/ Thorn_in_the_flesh | [1] en.wikipedia.org/wiki/ Thorn_in_the_flesh | [1] https://en.wikipedia.org/ wiki/Thorn_(letter) |
| Thorn in the flesh | Thorn in the flesh | Thorn in the flesh | Thorn (letter) |
| Thorn in the flesh is a phrase of New Testament origin used to describe a chronic infirmity, annoyance, or trouble in one's life, drawn from Paul the Apostle's use of the phrase in his Second Epistle to the Corinthians 12 : 7 -- 9 | Thorn in the flesh is a phrase of New Testament origin used to describe a chronic infirmity, annoyance, or trouble in one's life, drawn from Paul the Apostle's use of the phrase in his Second Epistle to the Corinthians 12 : 7 -- 9 | Thorn in the flesh is a phrase of New Testament origin used to describe a chronic infirmity, annoyance, or trouble in one's life, drawn from John the Apostle's use of the phrase in his Second Epistle to the Corinthians 12 : 7 -- 9 | Thorn or þorn (Þ, þ) is a letter in the Old English, Old Norse, Old Swedish and modern Icelandic alphabets, as well as modern transliterations of the Gothic alphabet, Middle Scots, and some dialects of Middle English. It was also used ... |
| **(A): Attributable** | **(B): Contradictory** | **(C): Contradictory** | **(D): Extrapolatory** |

Figure 2: Examples simulated from open-domain QA. We 1) use the original (question, answer, context) pair as an *attributable* instance (**A**), 2) substitute the answer or the answer span in the context to simulate a *contradictory* error example (**B, C**), and 3) replace the context with alternatives to simulate an *extrapolatory* error example (**D**). In order for models trained the simulated data to generalize well to the long answer setting in real-life search engines like New Bing, we convert the short answer to a long one (using ChatGPT).

objective of checking the supporting relationship between reference documents and generated statements. Summarization datasets, especially those involving the detection of hallucinations (including both intrinsic and extrinsic (Maynez et al., 2020), could provide a useful starting point for identifying attribution inconsistencies. Nevertheless, these datasets would require suitable adaptation. We keep their original data sequences and modify their data label space to suit the specific needs of the attribution evaluation definition. Additional information on this can be found in Appendix A.

**Simulate data from open-domain QA**. QA benchmarks provide an ideal platform for data simulation, as they comprise questions, their corresponding ground truth answers, and reference contexts. These elements can be directly employed as *attributable* examples (Figure 2, **A**). In open-domain QA datasets, answers are typically brief text spans. To cater to the long answer setting in most attributed LLMs, we convert these short answers into longer sentences using ChatGPT. For simulating *contradictory* errors, we propose two methods: (1) The first involves modifying the correct answer with an alternative candidate from an off-the-shelf QA model, an answer substitution model, or a random span generator (Figure 2, **B**). (2) The second retains the original answer but replaces the answer span in the reference context with a comparable candidate (Figure 2, **C**). To emulate *extrapolatory* errors, we employ a BM25 retriever on the ques-

tion, retrieving relevant external documents from resources such as Wikipedia, which do not contain the ground truth answers (Figure 2, **D**). More details regarding the simulation of these errors from QA datasets can be found in Appendix A.

## 4 Experimental Setup

### 4.1 Datasets

This section presents the datasets utilized for training and testing methods for automatic attribution evaluation. In particular, we develop two evaluation sets, *AttrEval-Simulation* and *AttrEval-GenSearch*, derived from existing QA datasets and a generative search engine, respectively. The dataset statistics are presented in Table 1.

**Training data**. To repurpose and simulate training examples, we follow the method in Section 3.2 based on four similar tasks' datasets. For *QA*, we consider NaturalQuestions (Kwiatkowski et al., 2019). For *fact-checking*, we include FEVER (Thorne et al., 2018), Adversarial FEVER (Thorne et al., 2019), FEVEROUS (Aly et al., 2021), VITAMINC (Schuster et al., 2021), MultiFC (Augenstein et al., 2019), PubHealth (Kotonya and Toni, 2020), and SciFact (Wadden et al., 2020). For *NLI*, we include SNLI (Bowman et al., 2015), MultiNLI (Williams et al., 2018), ANLI (Nie et al., 2020) and SciTail (Khot et al., 2018). For *summarization*, we include XSum-Halluc. (Maynez et al., 2020), XENT (Cao et al., 2022), and FactCC (Kryscinski

| Split | Related Tasks | Data Sources | #Samples |
|---|---|---|---|
| Train | QA | NaturalQuestions | 20K |
| | Fact-checking | FEVER, VITAMINC, Adversarial FEVER, FEVEROUS, SciFact PubHealth, MultiFC | 20K |
| | NLI | SNLI, MultiNLI ANLI, SciTail | 20K |
| | Summarization | XSum-Hallucinations, XENT, FactCC | 3.8K |
| Test | QA | PopQA, EntityQuestions, HotpotQA, TriviaQA, WebQuestions, TREC | 4K |
| | - | Annotated samples from a generative search engine | 242 |

Table 1: Statistics of the training and test datasets for attribution evaluation. We include the distributions of the labels and data sources in Appendix B.

et al., 2020). We use all examples in the summarization task datasets, and sample 20K examples from QA, fact-checking, and NLI task datasets. We combine all the simulated datasets to create the training set for our main experiment.

**AttrEval-Simulation.** For testing, we first simulate examples from six out-of-domain QA datasets: HotpotQA (Yang et al., 2018), EntityQuestions (Sciavolino et al., 2021), PopQA (Mallen et al., 2022), TREC (Baudis and Sedivý, 2015), TriviaQA (Joshi et al., 2017), and WebQuestions (Berant et al., 2013). Note that we intend to use different QA datasets for training and testing, as to test the model's generalization ability, and evaluate its performance across a diverse set of domains and question formats. Our manual examination indicates that 84% of 50 randomly sampled examples accurately align with their category, and the labeling errors are primarily due to incorrect annotations in the original QA datasets or heuristics used to formulate comparable answer candidates for contradictory errors and to retrieve negative passages for extrapolatory errors.

**AttrEval-GenSearch.** To examine the real-life application of automatic attribution evaluation, approximately 250 examples from the New Bing search engine are annotated carefully by the authors. This process comprises two subtasks: creating queries and verifying attributions. To avoid the issue of training data contamination, new queries are manually created across 12 domains (Figure 3).[6] To facilitate and motivate query annotation,

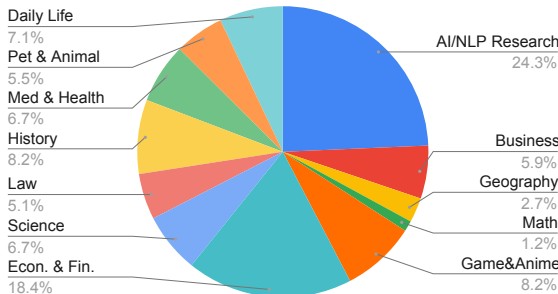

Figure 3: Domain distribution of our annotated AttrEval-GenSearch test set (covering 12 domains in total).

keywords from a specific domain are randomly generated using ChatGPT, and relevant facts within that domain are compiled from the Web.[7]

In the verification process, queries are sent to the New Bing search engine under a balanced mode following Liu et al. (2023), which balances accuracy and creativity. The validity of the output generated by New Bing is evaluated, where we consider only the first sentence that answers the question along with its reference. As we state in Section 2, our evaluation emphasizes the error type in a single reference per statement. In the case of a sentence having multiple references or distinct segments (for example, "XXX [1][2]" or "XXX [1] and YYY [2]"), each reference or segment is treated as a separate sample, and the attributions are verified individually. Finally, the samples are categorized by the annotators as attributable, contradictory, or extrapolatory. Detailed annotation guidelines can be found in Appendix D.

### 4.2 Implementation Details

In the configuration of "prompting LLMs", we test Alpaca (Taori et al., 2023), Vicuna (Chiang et al., 2023), ChatGPT (OpenAI, 2023a) and GPT-4 (OpenAI, 2023b), where we use OpenAI's official APIs (gpt-3.5-turbo, gpt-4-0314)[8], and weights from Alpaca and Vicuna from the official repository[9]. For Alpaca and Vicuna inference, documents are tokenized and truncated at a maximum of 2048 tokens. We generate text with a temperature of 0. The prompts for the task of evaluating attribution are provided in Appendix Table 6,

---

[6]The "AI/NLP Research" domain is inspired by recent discussions on social media about testing LLMs' knowledge on researchers, e.g., *"Is XX a co-author of the paper XX?"*

[7]We make an effort to collect new facts post-2021 to test about "knowledge confliction" (Zhou et al., 2023; Xie et al., 2023) between parametric and external knowledge.

[8]platform.openai.com/docs/api-reference/chat. Given GPT-4's high cost and slow inference speed, we evaluate it on 500 random samples from AttrEval-Simulation.

[9]https://github.com/tatsu-lab/stanford_alpaca, https://github.com/lm-sys/FastChat

| Setting | Model (Size) | AttrEval-Simulation | | | | AttrEval-GenSearch | | | |
|---|---|---|---|---|---|---|---|---|---|
| | | Attri. | Contra. | Extra. | Overall | Attr. | Contra. | Extra. | Overall |
| Zero-shot | Alpaca (7B) | 50.0 | 4.0 | 1.4 | 33.6 | 50.7 | 8.6 | 3.6 | 34.3 |
| | Alpaca (13B) | 48.3 | 5.6 | 2.2 | 33.5 | 50.6 | 6.1 | 19.3 | 34.7 |
| | Vicuna (13B) | 46.3 | 8.3 | 21.6 | 34.6 | 54.4 | 13.3 | 26.1 | 41.4 |
| | ChatGPT | 45.7 | 17.9 | 52.7 | 43.2 | 61.2 | 20.6 | 53.3 | 55.0 |
| | GPT-4 | **58.7** | **23.2** | **61.5** | **55.6** | **87.3** | **45.0** | **89.6** | **85.1** |
| Few-shot | Alpaca (7B) | 45.4 | 8.2 | 9.6 | 31.9 | 49.6 | 5.2 | 13.5 | 37.2 |
| | Alpaca (13B) | 38.9 | 20.1 | 2.2 | 33.1 | 50.5 | 10.3 | 5.6 | 34.8 |
| | Vicuna (13B) | 35.4 | **37.2** | 0.3 | 32.6 | 50.6 | 9.1 | 8.4 | 34.1 |
| | ChatGPT | 46.6 | 27.6 | 35.8 | 39.2 | 62.6 | 26.8 | 49.5 | 53.3 |
| | GPT-4 | **61.1** | 31.3 | **68.8** | **60.0** | **85.2** | **53.3** | **88.9** | **84.3** |
| Fine-tuned | Roberta (330M) | 62.5 | 54.6 | 74.7 | 65.0 | 47.2 | 25.2 | 62.3 | 49.8 |
| | GPT2 (1.5B) | 63.6 | 54.6 | 71.9 | 63.5 | 51.1 | 18.6 | 60.7 | 47.4 |
| | T5 (770M) | 45.9 | **57.1** | 71.6 | 59.1 | 58.5 | 24.3 | 72.5 | 61.6 |
| | Flan-T5 (770M) | 57.3 | 50.1 | 70.5 | 59.3 | 64.3 | 27.6 | 72.9 | 64.5 |
| | Flan-T5 (3B) | 48.1 | 48.7 | 67.1 | 55.7 | 77.7 | **44.4** | **80.0** | **75.2** |
| | Flan-T5 (11B) | 48.4 | 49.9 | 66.5 | 55.4 | **81.6** | 38.9 | 76.9 | 72.7 |
| | LLaMA (7B) | 62.2 | 50.7 | 74.6 | 62.8 | 77.9 | 41.1 | 78.3 | 72.5 |
| | Alpaca (7B) | **66.8** | 41.1 | 76.8 | 64.5 | 73.0 | 30.2 | **80.0** | 72.5 |
| | Alpaca (13B) | 63.6 | 48.9 | 75.8 | 63.6 | 77.5 | 34.5 | 79.4 | 73.3 |
| | Vicuna (13B) | 66.2 | 49.1 | **78.6** | **66.0** | 69.4 | 37.7 | 79.9 | 72.1 |

Table 2: The performance (F1 score) of AttrScore with different models on AttrEval-Simulation and AttrEval-GenSearch sets. The best-performing result in each setting is in **bold**. The results show both promising signals and challenges (e.g., all models struggle with contradictory errors) in automatic evaluation of attribution.

and our main results are averaged over 4 different prompts. For the few-shot prompting setting, we manually write 3 examples as demonstrations for both test sets as shown in Table 7. If LLMs yield an attribution label with an explanation, we extract the predicted label with regular expression.

In the "fine-tuning LMs" setting, we fine-tune four types of LMs of various scales: Roberta (340M) (Liu et al., 2019), (FLAN-)T5 (770M, 3B, 11B) (Raffel et al., 2020; Chung et al., 2022), GPT2 (1.5B) (Radford et al., 2019), LLaMA (7B), Alpaca (7B, 11B) (Taori et al., 2023), and Vicuna (7B, 11B) (Chiang et al., 2023). Our implementation utilizes the Huggingface library (Wolf et al., 2019) and Alpaca examples. The training is performed on 4 A100 80GB GPUs with a maximum of 512 tokens. For the LLaMA family of models, we use a batch size of 32 and train for 1 epoch. For the other models, we use a batch size of 64 and 3 epochs. We set the learning rate as 2e-5 and use a cosine learning rate decay with 0.03 warm-up steps.

**Metrics.** For evaluation, we present the F1 score for each individual class as well as the micro-F1 score, which is equivalent to the overall accuracy.

# 5 Results

## 5.1 Overall Performance

Table 2 presents an evaluation of different models on both the simulated dataset (AttrEval-Simulation) and the annotated dataset on New Bing (AttrEval-GenSearch). Our primary findings are as follows:

**GPT-4 achieves promising results,** reaching an overall accuracy of 81-83% on AttrEval-GenSearch and significantly outperforming other models. This suggests a promising potential for employing GPT-4 for automatic attribution evaluation to alleviate human annotation workloads, aligning with the emerging trend that uses GPT-4 for different evaluation tasks (Chiang et al., 2023; Zheng et al., 2023). However, it may still not be sufficiently accurate for practical use. We also note some potential concerns of bias (see Limitations Section 8).

**Automatic attribution evaluation presents substantial challenges.** This complex task requires not only understanding the reference information but also comparing it with the information in the statement, all of which can significantly vary across different datasets and test conditions. Against these challenges, models other than GPT-4 exhibit suboptimal performance in zero-shot and few-shot settings. Fine-tuning LMs on the simulated datasets from related tasks significantly improves the per-

| Error Type (percentage) | Example |
|---|---|
| Fine-grained information insensitivity (30.6%) | **Query:** Was ketchup a medicine before?
**Answer::** In the early 1800s , it was believed that ketchup could be used to treat indigestion, jaundice, and diarrhea
**Reference:** According to John Cook Bennett, tomatoes could be used to treat diarrhea, ... Soon, Bennett was publishing tomato ketchup recipes in pill form ... According to Dr. Bennett, this tomato blend can help people with diarrhea, indigestion, rheumatism, and jaundice.
▷ **Ground Truth:** Extrapolatory
▷ **GPT-4:** Attributable. The reference supports the claim that in the early 1800s , ketchup was believed to be used as a medicine to treat indigestion, jaundice, and diarrhea. It mentions John Cook Bennett's belief in the medicinal properties of tomatoes and his publication of tomato ketchup recipes in pill form, which were sold as patent medicines in supermarkets and pharmacies. |
| Misunderstanding the task definition and logical relation implied by labels (22.2%) | **Query:** When GTA6 will be published?
**Answer:** Rockstar Games has not officially announced the release date for GTA 6 yet.
**Reference:** The most likely GTA 6 release date is holiday 2024, and although there is no official confirmation, this estimate is confirmed by Microsoft's stating that it "is expected" in 2024 in a legal filing and assorted insider leaks.
▷ **Ground Truth:** Attributable
▷ **GPT-4:** Extrapolatory. The reference provides an estimated release date for GTA 6 based on a legal filing from Microsoft and insider leaks, but it does not confirm an official announcement from Rockstar Games regarding the release date. |
| Failing on symbolic operators (13.9%) | **Query:** What's the average salary for a software engineer working at Amazon in 2022?
**Answer:** According to Indeed.com, the average salary for a software engineer working at Amazon in the United States is $131,930 per year.
**Reference:** Average salary $132,147 Salary estimated from 3,612 employees, users, and past and present job advertisements on Indeed in the past 12 months. Last updated: April 18, 2023.
▷ **Ground Truth:** Contradictory
▷ **GPT-4:** Attributable. The reference does provide an average salary of $132,147 for a software engineer working at Amazon, which is very close to the claim's figure of $131,930. |

Table 3: Qualitative analysis on misclassified examples from prompting GPT-4 in AttrScore.

formance. For instance, the Vicuna (13B) model sees the overall accuracy on the two test sets rise from 34.6% and 41.4% in the zero-shot setting to 66.0% and 71.3%, respectively. And the fine-tuned FLAN-T5 (770M) model can even surpass Chat-GPT on both test sets. Despite this, there is still a large room for further improvement. Some models that yielded better results on the simulated test set may be less effective on the annotated test set, indicating a lack of consistency across diverse testing settings, signaling generalizability challenges.

**Models struggle most notably with contradictory errors**. Detecting contradictions is particularly complex because it requires the model to weigh one piece of information in the statement against another in the reference, a process that necessitates advanced fine-grained information comparison and reasoning capabilities. Consequently, even the best-performing model GPT-4 and the fine-tuned models often fail when faced with contradictory inputs, most often treating them as attributable (see qualitative analysis in Section 5.2).

### 5.2 Qualitative Analysis

To shed light on the space for future improvements in attribution evaluation, we qualitatively examine all the error examples of GPT-4 in the zero-shot setting. Representative examples are in Table 3.

Our first observation is that a significant portion (30.6%) of errors happen due to fine-grained information insensitivity: failure in comparing very fine-grained information such as numerical values,

numbers, dates, and time. Besides, the model misunderstands task definition and misinterprets logical relations implied by labels (22.2%). The model also struggles with symbolic operators (13.9%). For example, it fails to distinguish 'equal to' ($=$) and 'approximately equal to' ($\approx$) in numeric comparisons. In the left cases, the model tends to overlook the context clues and does not make judgments by conditioning on the reference (e.g., potentially relying on its own parametric knowledge).

Our observations point to two potential directions for improvement: 1) training or prompting models to be more faithful and strongly conditioned on the reference (Zhou et al., 2023), especially paying attention to fine-grained information; and 2) augmenting an LM-based evaluation method with external tools for different types of numerical and logical operations that are hard to be accurately performed only by the LM itself (Chen et al., 2020; Mialon et al., 2023). Similarly, we do qualitative analysis for ChatGPT in Appendix Section E.

### 5.3 Ablation Study

In this section, we perform an ablation study to test how each task influences the fine-tuned LMs' results and analyze the prompt sensitivity in zero-shot and few-shot settings for prompting LLMs.

**Contribution of individual task.** We show the performance of models fine-tuned on individual task datasets and their combinations in Figure 4. We select a representative from each group of the models under the fine-tuned setting in Table 2. Our findings

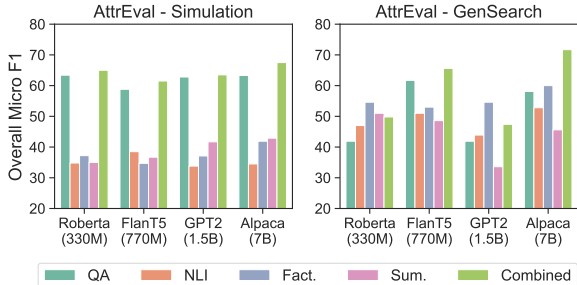

Figure 4: The influence of individual task data. Combining datasets generally improves model performance.

| Models | Task Prompts | Zero-shot | | Few-shot | |
|---|---|---|---|---|---|
| | | Sim. | Gen. | Sim. | Gen. |
| Alpaca | Attr. | 34.8 | 34.4 | 31.3 | 33.5 |
| | NLI | 32.1 | 35.5 | 32.1 | 33.6 |
| | Fact. | 34.0 | 33.9 | 32.7 | 46.7 |
| | Sum. | 33.6 | 33.5 | 31.6 | 34.8 |
| | Average | 33.6 | 34.3 | 31.9 | 37.2 |
| ChatGPT | Attr. | 37.2 | 45.1 | 37.6 | 51.4 |
| | NLI | 45.0 | 61.7 | 35.8 | 56.1 |
| | Fact. | 44.8 | 54.9 | 43.2 | 54.9 |
| | Sum. | 45.6 | 58.1 | 40.2 | 50.6 |
| | Average | 43.2 | 55.0 | 39.2 | 53.3 |

Table 4: Sensitivity of prompts for prompting LLMs on AttrEval-Simulation (Sim.) and -GenSearch (Gen.). The prompts include a prompt for attribution (Attri.), a NLI prompt, a fact-checking prompt (Fact.), and a summarization hallucination detection prompt (Sum.).

suggest that examples from our simulated QA and fact-checking task most significantly improve performance for the attribution evaluation task, hinting at a strong link between these tasks. Furthermore, integrating various related task datasets generally leads to better performance, particularly on out-of-domain test instances in AttrEval-GenSearch.

**Sensitivity of prompts.** The choice of prompts used to evaluate language models can have an impact on their performance. We evaluate the sensitivity of prompts for AttrScore under both zero-shot and few-shot settings of Alpaca (7B) and ChatGPT. We show four types of prompts as mentioned earlier: a prompt designed specifically for our evaluation setting (Attri.), an NLI prompt, a fact-checking prompt (Fact.), and a summarization hallucination detection prompt (Sum.). These prompts are presented in Appendix Table 6. As shown in Table 4, fact-checking and NLI prompts generally perform better, as similar tasks may have been seen during their instruction tuning phase.

## 6 Related Work

**Attributed LMs.** Generative LMs often produce hallucinations (Maynez et al., 2020; Dziri et al., 2021; Lee et al., 2018; Shuster et al., 2021; Wang and Sennrich, 2020; Xiao and Wang, 2021; Ji et al., 2023). To alleviate the issue, recent work proposes to augment LLMs (Mialon et al., 2023) with external tools (Schick et al., 2023; Li et al., 2023; Qin et al., 2023) such as retrievers (Guu et al., 2020; Lewis et al., 2020; Shuster et al., 2021; Izacard and Grave, 2021; Izacard et al., 2022; Borgeaud et al., 2022; Trivedi et al., 2022; Qian et al., 2023) and search engines (Nakano et al., 2021; Komeili et al., 2022; Thoppilan et al., 2022; Yao et al., 2022; Glaese et al., 2022; Shuster et al., 2022; Peng et al., 2023). Incorporating external references for generation inherently implies that the generated statement is backed by these references. However, the

validity of such attribution remains questionable.

**Evaluation of attribution.** To evaluate attribution, Liu et al. (2023) conduct a human evaluation to audit the verifiability of responses from generative search engines. They find that these engines frequently contain unsupported statements and inaccurate citations, which strengthen the need to carefully examine the attribution of generations (Rashkin et al., 2021). However, human evaluations are very expensive and time-consuming. Gao et al. (2022); Bohnet et al. (2022); Gao et al. (2023) propose to automatically evaluate attribution by levering NLI models (Honovich et al., 2022; Kamoi et al., 2023; Gekhman et al., 2023). We study this problem in a more comprehensive and realistic manner: 1) we explore how helpful other relevant tasks besides NLI are to attribution evaluation; 2) our evaluation setting is based on both benchmark examples and real examples.

## 7 Conclusion

In this paper, we investigate the important problem of automatically evaluating attribution given by LLMs. We begin by defining different types of attribution errors and then explore two approaches for automatic evaluation: prompting LLMs and fine-tuning smaller LMs. We experiment with both simulated test examples and manually curated test examples from a real-life generative search engine. The results highlight both promising signals and remaining challenges for the automatic evaluation of attribution. We hope our work could lay the foundation for future studies on this important problem.

## 8 Limitations

Currently, smaller models in AttrScore are fine-tuned on the combination of simulated or repurposed datasets from related tasks. However, this dataset still has gaps from the real scenario. Moreover, the error patterns in these simulated datasets might be overly simplistic and lack diversity, which can limit the models' ability to effectively handle more complex and varied real-world errors. It is also worth noting that these simulated datasets may contain noise and erroneous labels, which could further impede the models' learning and subsequent performance. How to obtain higher-quality training data for attribution evaluation at scale can be a major focus for future development.

Our annotated evaluation set, AttrEval-GenSearch, is derived from New Bing, which uses GPT-4 as its backbone. It is crucial to note that we also use GPT-4 for evaluating attribution on AttrEval-GenSearch, which achieves the best performance with around 85% overall accuracy. Some bias might come from GPT-4 both generating the test examples and evaluating the attribution, which could potentially skew our understanding of the model's true performance. We therefore caution against over-optimism. We also acknowledge that the size of AttrEval-GenSearch is moderate, which may not fully represent the real use setting of attributed LLMs.

While acknowledging current limitations, several promising directions emerge for future research and enhancement. For example, one can diversify data sources to include examples from a variety of generative search engines, not just New Bing. In addition, it may be beneficial to annotate larger-scale queries that cover a broad spectrum of topics, styles, and perspectives.

## 9 Ethics Statement

This research project involves evaluating attribution given by attributed LLMs. We collect and annotate data for evaluation using publicly available information on the web, with the assistance of a generative search engine, New Bing. We acknowledge that LLMs have the potential to reproduce and amplify harmful information present in the data. We made an effort to mitigate this risk by carefully selecting our evaluation data and by conducting analyses to identify and mitigate potential risks in the process.

## Acknowledgements

The authors would like to thank the anonymous reviewers and colleagues from the OSU NLP group for their thoughtful comments. This research was supported in part by NSF IIS 1815674, NSF CAREER 1942980, NSF OAC-2112606, and Ohio Supercomputer Center (OSC, 1987). The views and conclusions contained herein are those of the authors and should not be interpreted as representing the official policies, either expressed or implied, of the U.S. government. The U.S. Government is authorized to reproduce and distribute reprints for Government purposes notwithstanding any copyright notice herein.

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

# A Data Simulation

## A.1 Simulation - QA

**Attributable.** Since we have questions, and their ground truth answers and reference contexts, we can directly treat them as "Attributable" examples.

**Contradictory.** To simulate contradictory errors, we consider two methods. The first method involves modifying the correct answer by replacing it with a different candidate generated from an off-the-shelf QA model, an answer substitution model, or a random span generator. The second method involves keeping the original answer and replacing the answer span in the reference context with a similar candidate. The QA model, the answer substitution model, and the random span generator are all implemented by prompting a FLAN-T5-XL (3B) (Chung et al., 2022) with different task prompts in Appendix Table 5.

**Extrapolatory.** To simulate extrapolatory errors, we employ a BM25 retriever to retrieve external documents that do not contain ground truth answers from knowledge sources like Wikipedia or the Web. And then we replace the original paragraph with one of the retrieved documents. For the answer, we either keep the original ground truth answer or leverage a QA model to generate an answer. Here are more details for constructing negative retrieved documents in each dataset.

Following previous work (Karpukhin et al., 2020), we utilize the passages from Wikipedia dumps for constructing evidence for NaturalQuestions (Kwiatkowski et al., 2019), WebQuestions (Berant et al., 2013), and TREC (Baudis and Sedivý, 2015) datasets. In particular, we regard the highest-ranked passage including answers from BM25 as positive evidence and the top passage without answers as negative evidence.

For TriviaQA (Joshi et al., 2017), we select the passage with the highest overlap with answers from web texts as positive evidence and the top-ranked wiki passage without answers from BM25 as negative evidence. We exclude examples where the positive evidence has an overlap ratio of less than 0.5 with answers. For HotpotQA (Yang et al., 2018), we combine the ground truth passages provided as positive evidence and randomly select two out of eight passages provided as negative evidence. Similarly, in PopQA (Mallen et al., 2022), we find positive evidence from Wikipedia content through the provided link and retrieve negative evidence from Wikipedia dumps using BM25. In EntityQuestions (Sciavolino et al., 2021), we match positive evidence in Wikipedia texts searched by the question entity and retrieve negative evidence via BM25.

**Converting short answers to long sentences.** Since many of the attributed LLMs generate long sentences to the query, to make it our simulated data more realistic, we convert short answers to long answers using ChatGPT. Specifically, we prompt ChatGPT with the instruction *"Convert a given question and answer pair into plain sentences. [Question] [Answer]"*.

## A.2 Simulation - Fact Checking

With provided Wiki content as evidence in FEVER (Thorne et al., 2018) and Adversarial FEVER datasets (Thorne et al., 2019), we repurpose 'SUPPORTS' examples as attributable, 'REFUTES' as contradictory, and 'NOT ENOUGH INFO' as extrapolatory. Using the same label mapping, we apply this approach to the claim and evidence provided in VITAMINC (Schuster et al., 2021), after removing duplicated examples as shown in FEVER. For FEVEROUS (Aly et al., 2021), we concatenate all pieces of evidence, including tables and texts, and prepend an increasing index as the final evidence. We then ground the label into our three categories using the same label mapping. Regarding natural claim datasets with various label spaces, we keep the top 6 classes out of 117 in MultiFC (Augenstein et al., 2019) and map them to our defined three categories. In PUBHEALTH (Kotonya and Toni, 2020), we consider both 'unproven' and 'mixture' classes as extraplanetary. We also regard the abstract of the article as evidence. For SciFact (Wadden et al., 2020), we repurpose 'SUPPORT' as attributable and 'CONTRADICT' as contradictory. Additionally, we randomly select one sentence from the abstract of other articles as evidence for the 'Not enough information' class to construct extrapolatory examples.

## A.3 Simulation - NLI

Natural language inference (NLI) aims to determine whether a hypothesis is true given a premise. In NLI datasets such as SNLI (Bowman et al., 2015), MultiNLI (Williams et al., 2018), ANLI (Nie et al., 2020), and SciTail (Khot et al.,

2018), the hypothesis is considered the claim and the premise is regarded as the evidence. The original labels in NLI datasets, namely 'Entailment', 'Contradictory', and 'Neutral', are mapped to 'Attributable', 'Contradictory', and 'Extrapolatory'.

### A.4 Simulation - Summarization

Summarization involves condensing a given passage or article into brief sentences while preserving its original meaning. To simulate contradictory examples, we use datasets with annotations of hallucinations. In terms of XSum-Hallucination (Maynez et al., 2020), we merge examples with the same ID and consider those with the most intrinsic hallucination as contradictory and those with the most extrinsic hallucination as extrapolatory. Paired full articles and ground truth summaries are treated as attributable examples. For XENT (Cao et al., 2022), 'Non-factual Hallucination' and 'Intrinsic Hallucination' are seen as contradictory, 'Factual Hallucination' as extrapolatory, and 'Non-hallucinated' as attributable. Each article and reference are paired as attributable examples. Finally, we resplit the manually annotated dev and test sets for training and evaluation in FactCC (Kryscinski et al., 2020), with 'INCORRECT' labeled as extrapolatory and 'CORRECT' as attributable.

### B  Label and Subset Distributions of Training and Test Sets

We show the label and data sources' distributions of training and AttrEval-Simulation sets in Figure 5 and Figure 6.

### C  Prompts for LLMs as AttrScore

We show different kinds of prompts for using LLMs as AttrScore in Table 6. And we show the few-shot demonstrations in Table 7.

### D  Generative Search Engine Examples Annotation Protocol

We show the detailed annotation guidelines in the following.

# Annotation Guidelines

## Overview

Thank you for participating in this annotation task. The goal of this task is to create a query and verify whether a given reference document fully supports the generation of the query.

There are two sub-annotation tasks:
1. Create a query based on a few given keywords under a topic.
2. Verify whether a given answer to a query is fully supported by its references.

## Task 1: Create a query for a specific domain.

You will be shown a list of keywords (e.g., inflation rate, CPI, GDP, unemployment rate, etc.) from a specific domain or topic (e.g., economics) and a demo question (e.g., What was the unemployment rate in Germany in 2020?) as an inspiration. Then you will be asked to create a new query based on these keywords.

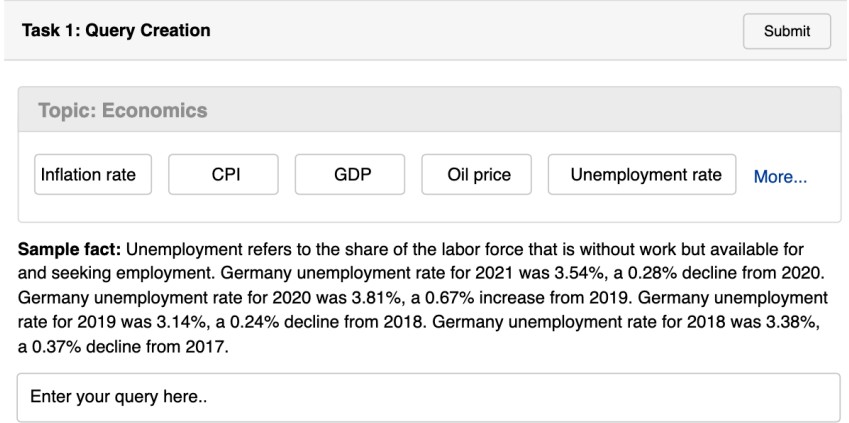

## Task 2: Verify whether the generated statement is supported by its reference.

You will be shown a user query, a generative search engine's response, and associated references. You will need to read the query, response, and reference carefully and verify whether the cited evidence fully supports the generation of the query.

You will be given three options:
a. **"Attributable"**: The reference fully supports the generation of the query.
b. **"Extrapolatory"**: The generation contradicts the reference.
c. **"Contradictory"**: The reference does not contain sufficient information to infer the correctness of the generation.

Consider the following examples:
(1) "Attributable"
**Query:** What is the temperature range on the moon?
**Response:** The average temperature on the moon can range from -298 degrees F (-183 degrees C) at night to 224 degrees F (106 degrees C) during the day [1].
**Reference:** The average temperature on the Moon (at the equator and mid latitudes) varies from -298 degrees Fahrenheit (-183 degrees Celsius), at night, to 224 degrees Fahrenheit (106 degrees Celsius) during the day. Because the Moon has no...
**Annotation:** Attributable

(2) "Extrapolatory"
**Query:** What is the highest price of a gallon of regular gas in the US in 2022?
**Response:** According to a news article by NPR, as of June 2022, the highest average price of a gallon of regular gas in the US was $6.34 in California [1].
**Reference:** U.S. gas prices hit a new record on Monday, with a gallon of regular gas costing an average of $4.865 (not adjusting for inflation), according to the most recent data from the American Automobile Association...
**Annotation:** Extrapolatory

(3) "Contradictory"
**Query:** What was the unemployment rate in Germany in 2020?
**Response:** According to [1], the unemployment rate in Germany for 2020 was 4.31%, which was a 1.17% increase from 2019. However, it is important to note that the COVID-19 pandemic had a significant impact on unemployment rates in Germany and around the world.
**Reference:** Germany unemployment rate for 2020 was 3.81%.
**Annotation:** Contradictory

---

**Task 2: Verification**                                            Submit

---

Query: What was the unemployment rate in Germany in 2020?

Response: According to [1], the unemployment rate in Germany for 2020 was 4.31%, which was a 1.17% increase from 2019. However, it is important to note that the COVID-19 pandemic had a significant impact on unemployment rates in Germany and around the world.

Reference: [1] https://www.macrotrends.net/countries/DEU/germany/unemployment-rate
Germany unemployment rate for 2020 was 3.81%.

---

Does the evidence fully supports the response?       [          ▾ ]

                                                     Attributable

                                                     Extrapolatory

                                                     **Contradictory**

| Tasks | Prompts |
|---|---|
| QA | Context: [Context]\n
Based on Context, [Question] |
| Answer Substitution | Please provide a related term or substitution for the given input, which should be different from the input.\n
"Input: Biden; Output: Obama\n"
"Input: 1949; Output: 1358\n"
"Input: University of Maryland; Output: University of Cambridge\n"
"Input: 09/12/2014; Output: 03/30/2008\n"
"Input: $431; Output: $769;\n"
"Input: [Ground Truth Answer]; Output: ", |
| Random Span Generation | Extract a phrase from the given passage. \n Passage: [Context] |

Table 5: Prompts for QA, answer substitution, and random span generation when simulating contradictory errors

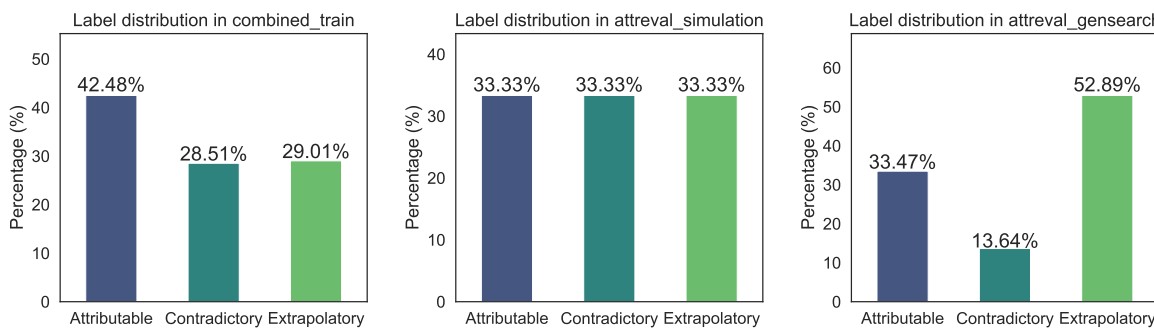

Figure 5: Label distribution of training and test sets.

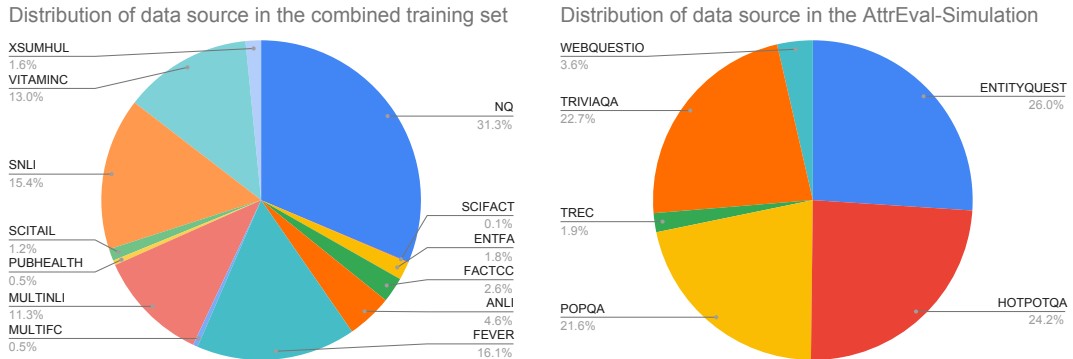

Figure 6: Data source distribution of combined training and AttrEval-Simulation sets.

## E  Additional Qualitative Analysis

The qualitative results of ChatGPT are shown in Table 8. Our first observation is that a significant portion (79.4%) of errors happen due to ChatGPT overlooking the context clues and does not make judgments by conditioning on the reference (e.g., potentially relying on its own parametric knowledge). For the remaining error cases, they are: 1) fine-grained information insensitivity (13.8%): failure in comparing very fine-grained information such as numerical values, numbers, dates, and time; 2) failure in performing symbolic operations (6.8%): the model fails to verify the claim which requires performing symbolic operations over the reference, such as verifying set relationships.

## F  Author Contribution Statement

Xiang Yue conceived the project, conceptualized and designed the study, conducted experiments, wrote the manuscript, and annotated New Bing test examples. Boshi Wang provided critical feedback and edits, revised the manuscript, contributed to the

| Prompt Types | Prompts |
|---|---|
| Attribution | ### Instruction:

As an Attribution Validator, your task is to verify whether a given context can support the claim. A claim can be either a plain sentence or a question followed by its answer. Specifically, your response should clearly indicate the relationship: Attributable, Contradictory or Extrapolatory. A contradictory error occurs when you can infer that the answer contradicts the fact presented in the context, while an extrapolatory error means that you cannot infer the correctness of the answer based on the information provided in the context.

### Input:
Claim: [Question Answer] or [Plain Sentence] \n\n
Context: [Context]

### Response: |
| Fact-Checking | ### Instruction:
Fact-check a claim based on the given evidence.
Options: Supported, Refuted or Not Enough Information

### Input:
Claim: <Claim>\n\n
Evidence: <Evidence>

### Response: |
| NLI | ### Instruction:
Read the following and determine if the hypothesis can be inferred from the premise.
Options: Entailment, Contradiction, or Neutral

### Input:
Hypothesis: <Hypothesis>\n\n
Premise: <Premise>

### Response: |
| Summarization Hallucination Detection | ### Instruction:
Read the following and determine whether the source text can support the summary.
Options: Support, Contradicts, or Not Enough Information

### Input:
Summary: <Summary>\n\n
Source: <Source>

### Response: |

Table 6: Prompt variations for test the sensitivity of different prompts on the results. We use the "Attribution" prompt for our main experiments as default as it achieves the best performance overall.

conceptualization of the study, conducted experiments for ChatGPT, and annotated New Bing test examples. Ziru Chen set up the training code base, annotated New Bing test examples, and conducted experiments for fine-tuning Roberta, Flan-T5, and GPT-2. Kai Zhang contributed to all the simulation data preprocessing, revised the manuscript, and annotated New Bing test examples. Yu Su and Huan Sun secured funding for the project, provided supervision and guidance throughout the study, contributed to the conceptualization and design of the study, and edited the whole manuscript. All authors approved the final version of the manuscript.

---
Few-shot demonstrations
---
Here are some demonstration examples for you.

### Input:
Claim: In what year was the writer of the opera "Mazeppa" born? The writer of the opera "Mazeppa" was born in 1840.

Reference: Mazeppa, properly Mazepa, is an opera in three acts (six scenes) by Pyotr Ilyich Tchaikovsky. The libretto was written by Victor Burenin and is based on Pushkin\'s poem "Poltava". Pyotr Ilyich Tchaikovsky (25 April/7 May 1840 – 25 October/6 November 1893), often anglicized as Peter Ilich Tchaikovsky, was a Russian composer of the romantic period, some of whose works are among the most popular music in the classical repertoire. He was the first Russian composer whose music made a lasting impression internationally, bolstered by his appearances as a guest conductor in Europe and the United States. Tchaikovsky was honored in 1884, by Emperor Alexander III, and awarded a lifetime pension.

### Response:
Attributable. From the given reference, Pyotr Ilyich Tchaikovsky was the writer of "Mazeppa", who was born in 1840 as stated in the given claim.

-

### Input:
Claim: In what part of Buenos Aires Province is the city that has The Juan Carlos Castagnino Municipal Museum of Art located? It is located in the southeast part.

Reference: The Juan Carlos Castagnino Municipal Museum of Art is a museum of fine arts in Mar del Plata, Argentina. Mar del Plata is an Argentine city in the east part of Buenos Aires Province located on the coast of the Atlantic Ocean. It is the head of General Pueyrredón Partido. Mar del Plata is the second largest city in Buenos Aires Province. The name "Mar del Plata" has the meaning of "sea of the Silver region" or "adjoining sea to the (River) Silver region". Mar del Plata is one of the major fishing ports and the biggest seaside beach resort in Argentina.

### Response:
Contradictory. The reference says that Mar del Plata is in the east part of Buenos Aires, so The Juan Carlos Castagnino Municipal Museum of Art is also in the east part of Buenos Aires, not southeast.

-

### Input:
Claim: In what year was Louis Joseph de Frances\'s mother born? Louis Joseph de Frances's mother was born in 1755.

Reference: Lakshman Joseph de Saram is a film composer and classical musician. Born in Colombo, Sri Lanka and educated at the Royal College, Colombo, the High School of Performing Arts, Manhattan School of Music and Juilliard Pre-College in New York City, Joseph de Saram is influential in the music of South Asian art cinema, having scored many international award-winning films like \'Between Two Worlds\' and \'Akasa Kusum.\' His best-known score is to the 2012 film "Bel Ami". Joseph de Saram is also artistic director of The Chamber Music Society of Colombo. Louis Joseph de Lorraine "Duke of Guise" and Duke of Angoulême, (7 August 1650 – 30 July 1671) was the only son of Louis, Duke of Joyeuse and Marie Françoise de Valois, the only daughter of Louis-Emmanuel d\'Angoulême, Count of Alès, Governor of Provence and son of Charles de Valois Duke of Angoulême, a bastard of Charles IX of France.

### Response:
Extrapolatory. The answer to the question cannot be inferred from the given reference since the person "Louis Joseph de Frances" is not mentioned. We also cannot find any evidence in the Reference which contradicts with the provided answer.

-

Now here is the example that needs your response:
---

Table 7: Few-shot demonstrations for prompting LLMs as AttrScore.

| Error Type (percentage) | Example |
|---|---|
| Overlooking contextual cues (79.4%) | **Query:** Was ketchup a medicine before?
**Answer::** Yes, that's true! Ketchup was sold as a remedy for indigestion, diarrhea, jaundice and rheumatism in the 1830s
**Reference:** Decades later, in the 1870s, American entrepreneur Henry Heinz created a modern version of ketchup with ripe tomatoes, distilled vinegar, brown sugar, salt and a variety of spices. By 1905, the company had sold five million bottles of ketchup, cementing its place as the condiment we know and love today.
**Ground Truth:** Extrapolatory
**ChatGPT:** Attributable. The reference supports the claim that ketchup was sold as a remedy for various ailments in the 1830s, indicating that it was used as a medicine before. |
| Fine-grained information insensitivity (13.8%) | **Query:** How many total miles of coastline does Norway have?
**Answer:** The official length was recalculated in 2011 as 100,915 km, including fjords and islands.
**Reference:** A straight line along Norway's sea borders (the coastal perimeter) is 2,650 kilometers (1,650 mi) long. Along the coast there are many fjords, islands, and bays, resulting in a low-resolution coastline of over 25,000 kilometers (16,000 mi).
**Ground Truth:** Contradictory
**ChatGPT:** Attributable. The reference provides information that the official length of Norway's coastline, including fjords and islands, is 100,915 km. Therefore, the claim that Norway has a total of 100,915 km of coastline is supported by the Reference. |
| Failing on symbolic operators (6.8%) | **Query:** Is Andrew Ng an author of the recent paper "generative agents: interactive simulacra of human behavior"?
**Answer:** Yes, Andrew Ng is one of the authors of the paper "Generative agents: Interactive simulacra of human behavior".
**Reference:** Computer Science > Human-Computer Interaction [Submitted on 7 Apr 2023] Generative Agents: Interactive Simulacra of Human Behavior Joon Sung Park, Joseph C. O'Brien, Carrie J. Cai, Meredith Ringel Morris, Percy Liang, Michael S. Bernstein.
**Ground Truth:** Contradictory
**ChatGPT:** Attributable. The reference clearly states that Andrew Ng is one of the authors of the paper "Generative Agents: Interactive Simulacra of Human Behavior", which supports the claim that he is an author of the paper. |

Table 8: Qualitative analysis on misclassified examples from prompting ChatGPT in AttrScore.