# OpenReview forum: "Automatic Evaluation of Attribution by Large Language Models"
_EMNLP/2023/Conference — EMNLP 2023 Findings_

### Official Review · Reviewer_DLKU · 2023-08-04

**Soundness:** 3

**Excitement:**

3: Ambivalent: It has merits (e.g., it reports state-of-the-art results, the idea is nice), but there are key weaknesses (e.g., it describes incremental work), and it can significantly benefit from another round of revision. However, I won't object to accepting it if my co-reviewers champion it.

**Paper Topic And Main Contributions:**

This paper presents an automatic method to evaluate LLM's attributions, which link the generated text to the evidence texts. Automatic attribution evaluation is treated as a three-way classification task, with labels being: attributable, extrapolatory, and contradictory. The authors evaluated several different models for automatic attribution evaluation. These models include fine-tuning the LLMs on QA, Fact-checking, NLI, and Summarization datasets, as well as zero-shot or few-shot LLMs.

The models were evaluated on two test sets: one comprised of simulated data built from QA datasets, and the other collected from answers and documents generated by New Bing. The experiments demonstrated varying performances across different models.

**Questions For The Authors:**

A. Regarding evaluations of fine-tuned models on the AttrEval-Simulation dataset, is it technically similar to standard supervised training evaluations? While training and testing differ in that they use distinct datasets (as per line 286), their methods for generating extrapolatory and contradictory examples seem to be similar. This similarity could limit the implications of such evaluations.

**Reasons To Accept:**

1.	This paper demonstrates how closely the automatic attribution evaluation aligns with human evaluation.
2.	The research shows that models trained on repurposed (simulated) datasets result in improvements in the AttrEval-GenSearch dataset, which comes from actual generative models.

**Reasons To Reject:**

1.	The evaluation on 'AttrEval-Simulation' might be biased or may exhibit limited diversity due to the constrained ways of generating each label. This might not truly represent the types or the degree of challenges in evaluating attributions for LLM-generated answers.
2.	Evaluation by a model can be limiting when the evaluated model (LLMs which generate answers and attributions) outperforms the evaluating model. The evaluating model might penalize the evaluated model even if the latter's attribution is correct.

**Reproducibility:**

4: Could mostly reproduce the results, but there may be some variation because of sample variance or minor variations in their interpretation of the protocol or method.

**Reviewer Confidence:**

2: Willing to defend my evaluation, but it is fairly likely that I missed some details, didn't understand some central points, or can't be sure about the novelty of the work.

**Typos Grammar Style And Presentation Improvements:**

Footnote 5: It's unclear how the authors differentiate between “verifiability” and “ ’relevance’, i.e., whether to answer correctly responds to the query”.
Line226 : This paragraph describes how the ‘contradictory’ and ‘extrapolatory’ examples are generated. I am confused if this data was only used for the fine-tuning of the LLM or if it was also employed in the evaluation for ‘AttrEval – Simulation’.
Line 309: Does the term ‘verification process’ refer back to what's mentioned in Line 301? It was once lost in connecting it.

---

> ### Author Rebuttal · Authors · 2023-08-29
>
> Thank you for your insightful review and constructive feedback on our work. We appreciate your acknowledgment of our effective evaluation approaches and our exploration of repurposed and simulated datasets for fine-tuning smaller models.
>
> ---
>
> **Bias in AttrEval-Simulation:**
> We understand your concern about potential bias in the 'AttrEval-Simulation' dataset. While we acknowledge its limitations in the paper (Line 537-556), we believe it provides a valuable starting point for understanding the strengths and weaknesses of our framework. Our **simulation methods are inspired by real use cases in generative search engines**, providing a valuable proxy for testing our framework. Please also see our response to Reviewer XrPK under **Evaluation on Simulated Data**.
>
> ---
>
> **Limitation of the evaluating model:**
> This is a good point! However, practical considerations like efficiency and budget often restrict the use of large language models like GPT-4 for the evaluation. In such scenarios, our results show that **fine-tuned smaller models can still achieve reasonably good performance** (>70% accuracy) and even outperform ChatGPT in many cases, despite being trained on simulated datasets. We note that in the future if large-scale realistic datasets for training are available, the performance of these smaller models could be higher.
>
> ---
>
> **Extrapolatory class is an inappropriate category:**
> The decision to use a three-way classification reflects our aim to provide a nuanced understanding of attribution errors. **If a reference is “relevant” but does not directly contradict or support the generated statement, we treat this example as an “extrapolatory” error**. This is particularly important in high-stakes domains like healthcare and finance, where seemingly correct but unsupported statements can lead to significant risks.
>
> Regarding “the LLMs might be providing the correct or an appropriate answer due to their inherent generalization abilities,” that’s true, but still, attributable LLMs should always avoid generating statements without supporting references, which is what our extrapolatory class captures.
>
> ---
>
> **Questions A. Fine-tuning models on the AttrEval-Simulation dataset:**
> You made a very good point! Our simulation is designed with model generalizability in mind from the outset. We ensure this by using different QA datasets for training and testing simulations (Table 1; Line 279-289). Although the methods for generating 'extrapolatory' and 'contradictory' examples are similar in both datasets, the **actual data instances used for the simulation come from different sources**. In fact, for the test set simulation, we use **six different QA datasets**. This diversity in sources ensures that our model faces a wide range of scenarios and discourages memorization of specific examples, thus promoting generalizability tests.
>
> Notably, even when trained on the simulated training set, the performance of the model on the simulated test set (AttrEval-Simulation) is not perfect, with the best-performing model achieving just 66% accuracy. This result further emphasizes the generalizability challenge of our AttrEval-Simulation dataset.
>
> ---
>
> **Typos, Grammar, Style, and Presentation Improvements**
>
> Footnote 5: We apologize for any confusion. By "verifiability", we mean whether the generated statement is backed by the cited reference, which is a similar concept to attribution and is used in prior work (Liu et al., 2023). By "relevance" we mean whether the answer (which may or may not be supported by the reference) correctly responds to the user's query. We will make them more clear in the revised version.
>
> Line 226: The simulation method described here was used for generating part of the fine-tuning data and evaluation data. However, as we stated in the response to Question A, the actual data instances used to simulate fine-tuning and evaluation sets are sampled from different QA datasets to ensure the generalizability test. We will revise the text for clarity.
>
> Line 309: ‘Verification process’ does refer to the process mentioned in Line 301. We will work to improve the clarity in the revision.

---

### Official Review · Reviewer_82hu · 2023-08-05

**Soundness:** 3

**Excitement:**

4: Strong: This paper deepens the understanding of some phenomenon or lowers the barriers to an existing research direction.

**Paper Topic And Main Contributions:**

The paper presents an approach to automatically verify if a statement generated by a LLM is supported by the cited reference. The authors formulate the problem as a classification task, wherein a <question, statement, reference> pair is labeled as “attributable”,”extrapolatory” or “contradictory”.

They study two main directions: fine-tuning open-source LMs, and prompting SOTA LLMs in a zero/few-shot setting. The data used for fine-tuning is repurposed from several representative corpora from the fields of QA, fact-checking, NLI and summarization. For evaluation two types of data are considered, which encompass 12 domains: samples derived from QA datasets and author annotated samples from the New Bing search engine.

The authors experiment using a comprehensive set of models and provide an interesting discussion of the results.


**Questions For The Authors:**

Question A: While I appreciate the inclusion of the ‘contradictory’ label, I am not sure why a label for partially supported claims was not considered. I am thinking, in particular, about sentences with multiple claims and the scenario where the same sentence contains multiple references.  How often does this occur throughout the dataset?

Question B: Why do you think the few-shot performance is so underwhelming for the Alpaca/Vicuna/ChatGPT models (which actually regress on average, compared to the zero-shot setting)?

Question C: Does the size of the inputs have an impact on the results?  We are told specifically that inputs for Alpaca and Vicuna are truncated to 2048 tokens but we know that ChatGPT and GPT-4 can accept longer inputs. Are there inputs that do not fit in the 2048 limit (or the 512 size used during fine-tuning)? If yes, this information should be presented (along with the sizes accepted by the versions of ChatGPT/GPT-4 used in the experiments).

Question D: I believe at least a few samples might be incorrectly parsed (check lines 22-30 and 38-47 in AttrEval-GenSearch.csv and the parsing code in prompt_chatgpt_gpt4). Also, is the sample starting on line 38 labelled correctly (two games are missing from the reference enumeration and the answer itself contains only 8 examples)?


**Reasons To Accept:**

The paper explores an important present-day problem. A large selection of models are investigated on this task. The authors explore the limitations of the strongest available model (GPT-4). Additionally, the value of the provided training dataset is reflected by the competitive performance of relatively small fine-tuned models (Roberta/GPT2), which out-perform significantly larger LLMs (Alpaca/Vicuna) in a zero/few-shot setting.

**Reasons To Reject:**

Given the (acknowledged) potential bias of using GPT-4 (which was used for both generation and evaluation) and the fact that it was only evaluated on a subset of the test data, I think it might have been interesting for the follow-up discussion to highlight more the results of the smaller models.


**Reproducibility:**

4: Could mostly reproduce the results, but there may be some variation because of sample variance or minor variations in their interpretation of the protocol or method.

**Reviewer Confidence:**

4: Quite sure. I tried to check the important points carefully. It's unlikely, though conceivable, that I missed something that should affect my ratings.

**Typos Grammar Style And Presentation Improvements:**

L361: equivelant
L1105: extraplanetary -> extrapolatory
L1154: “. And” -> and
Table 5 Description missing “.”
Table 6 description: “prompt variations for test the sensitivity”

I believe the information that GPT-4 was only tested on 500/4000 samples from the test set should be more prominent, perhaps in the description of Table 2 (rather than in a footnote).

Also, in Table 2 it may be informative to display the maximum context size (if the size of the inputs makes it relevant).

---

> ### Author Rebuttal · Authors · 2023-08-29
>
> Thank you for your thoughtful review and constructive feedback on our work! We appreciate your recognition of the significance of the attribution problem we study and the comprehensive evaluation methods we explore.
>
> ---
>
> **Potential Bias:**
> We acknowledge the potential bias of using GPT-4 for both generation and evaluation, as stated in the Limitation section (Lines 537-556). We agree that emphasizing the results of smaller fine-tuned models in the follow-up discussion would be beneficial. To address this, we will solve your comments in Question B and C (see more below). Moreover, we have now **tested the performance of GPT-4 on the full AttrEval-Simulation dataset**, achieving comparable results to the subset evaluation (54.5 vs. 55.6 overall accuracy).
>
> ---
>
> **Question A:**
> We apologize if our original categorization was unclear. We do consider partially supported claims, and their labels depend on whether the unsupported part contradicts the reference (“contradictory”) or is not mentioned in the reference (“extrapolatory”).  For example, in the last example of Table 3, the answer can be mostly supported by the reference except the key information “$131,930” is wrong. We treat such examples as "contradictory."
>
> We agree that sentences with multiple claims and references do appear in real practice. However, our task setting prioritizes one reference per statement, a unit task that more complex scenarios can be decomposed to (Line 125-130). We study such a setting as it forms the basis for dealing with multiple references or distinct segments.
>
> ---
>
> **Question B:**
> Your question regarding the underwhelming few-shot performance of Alpaca, Vicuna, and ChatGPT models is insightful. We hypothesize this is because few-shot methods are often very sensitive to the demonstrations provided, which is a known problem. We will try different demonstration selection strategies and provide more discussion about this in the revision.
>
> ---
>
> **Question C:**
> We appreciate the reviewer's query about the input sizes. While some examples in our datasets do exceed 512 or 2048 tokens, these instances form a relatively small percentage (e.g., 4.1% and 0.2% respectively for the training set). We will add such information in the revision. But we do agree that dealing with very long sequences remains a challenge for future work.
>
> ---
>
> **Question D:**
> We appreciate the reviewer's diligence in identifying these instances! They are correctly parsed in the csv file but wrongly in the browser (due to new line symbol “\n”). And the label in line 38 should be “extrapolatory” as two games are missing in the reference. We will thoroughly review all the examples and their labels again.
>
> ---
>
> **Typos, Grammar, Style, and Presentation:**
> We are grateful for the reviewer's keen eye in identifying typos and presentation-related issues. We will correct them in our revised manuscript to enhance clarity and readability. Additionally, we will report the performance of GPT-4 on the full set instead of subset as we stated in the previous response under **Potential bias**.

---

### Official Review · Reviewer_XrPK · 2023-08-10

**Soundness:** 3

**Excitement:**

3: Ambivalent: It has merits (e.g., it reports state-of-the-art results, the idea is nice), but there are key weaknesses (e.g., it describes incremental work), and it can significantly benefit from another round of revision. However, I won't object to accepting it if my co-reviewers champion it.

**Paper Topic And Main Contributions:**

The paper explores the problem of attribution with LLMs by defining the task as having the classes attributable, contradictory, and extrapolatory, and testing out whether finetuned and prompted language models can succeed at classification.

The paper proposes to repurpose data from related tasks (NLI, fact-checking, QA) through mapping samples to the proposed problem formulation.

Performance is benchmarked and analyzed on two datasets they contribute: a 4K sample simulated dataset using QA data and a 242 sample test set generated from New Bing. Contradictions are found to be a common error pattern.


**Questions For The Authors:**

A) How does the data format/structure in QA, fact-checking, NLI relate to the results shown in Fig. 4? What are the similarities to the references in New Bing dataset?


**Reasons To Accept:**

The paper addresses attribution in LLMs which is important and especially relevant to extend LLMs to further applications.

It is helpful for the community to understand the effectiveness of various strategies for tasks like attribution; this paper’s exploration of both prompting and finetuning language models, especially recent ones, is appreciated.

The paper notes parallels in attribution between various subtasks of NLP (NLI, fact-checking, QA); the premise of transferring data/knowledge from those tasks is creative. The experiment to gauge impact by dataset type on performance is interesting.


**Reasons To Reject:**

Novelty of problem formulation is slightly limited.
- Formulation of attribute errors is highly reminiscent of NLI categories (i.e. entailment, neutral, contradiction).
- NLI for attribution has been studied as noted in (Gao et al. 2022) and (Bohnet et al. 2022).

Evaluating on simulated data does not necessarily represent the type of attribution seen in practice with the paper’s use case (e.g. search engines).

The size of AttrEval-GenSearch is relatively small (~250 images), bringing into question the significance of the results extending past simulated data. Additionally, the dataset may have limited use for future researchers due to this size.

When repurposing data from SNLI, the premises are single sentences. However, in the proposed attribution application, the “premises” are likely to be full references. There seems to be a significant domain gap, which could explain the ineffectiveness of NLI data in Figure 4.


**Reproducibility:**

4: Could mostly reproduce the results, but there may be some variation because of sample variance or minor variations in their interpretation of the protocol or method.

**Reviewer Confidence:**

3: Pretty sure, but there's a chance I missed something. Although I have a good feel for this area in general, I did not carefully check the paper's details, e.g., the math, experimental design, or novelty.

**Typos Grammar Style And Presentation Improvements:**

More data repurposing details would be helpful in the Approach section rather than Appendix A.

Line 361 type - equivelant -> equivalent

---

> ### Author Rebuttal · Authors · 2023-08-29
>
> Thank you for your constructive feedback on our work. We appreciate your recognition of the significance of the attribution problem we study, the different evaluation methods we explore, and the connection of attribution evaluation to tasks like NLI, fact-checking, and QA.
>
> ---
>
> **Novelty of Problem Formulation:**
> Yes, our attribution error categories do share similarities with NLI categories, but it's important to distinguish between attribution evaluation and NLI tasks. NLI primarily concerns the “logical'' relationship between the premise and hypothesis, with the premise typically being very short. Conversely, attribution evaluation concentrates on verifying whether fine-grained information in the reference supports the generation, with the reference typically being a lengthy document.
>
> While Gao et al. (2022) and Bohnet et al. (2022) do frame attribution as a *binary* NLI classification task, we argue that this oversimplification misses the nuance of attribution. Our work proposes a more **fine-grained categorization**, enhancing safe system usage and providing insight for the development of error-specific correction mechanisms. Our experiments show that **NLI models do not perform as well on our two test sets, suggesting that NLI and attribution may require different handling.**
>
> ---
>
> **Domain Gap between SNLI and Attribution:**
> Yes, there is indeed a gap! We're really glad you brought this up, as it's a crucial aspect of our work. We recognized this gap early on, and **it's precisely why we didn't limit ourselves to just NLI datasets or off-the-shelf NLI models** which existing work mostly focused on. We took a more exploratory route, delving into other related tasks like fact-checking and summarization, and even going a step further to propose methods to simulate data from QA. Our study can provide more insights into how these relevant tasks can be transferred to attribution evaluation (thanks for noting that in the Reasons To Accept). For example, according to the results in Figure 4, open-domain QA and fact-checking share more similarities to the attribution task, and combining the four datasets leads to the best generalizability.
>
> ---
>
> **Evaluation on Simulated Data:**
> While we acknowledged in the Limitation section (Line 525-530) that the simulated data has gaps from the real scenario, it's worth noting that our **simulated data is designed to mimic real use cases** in generative search engines. Our **simulation methods are inspired by real examples**, thus providing a valuable proxy for testing our framework. For example, to simulate extrapolatory errors where generated statements contain additional information that cannot be verified in the reference, we employ a BM25 retriever on the question, retrieving relevant external documents, which do not contain the ground truth answers.
>
> ---
>
> **AttrEval-GenSearch Dataset Size:**
> We acknowledge the relatively small size of our AttrEval-GenSearch dataset in the Limitation section (Line 547-550). We would like to point out there are similarly small eval datasets that are widely used in evaluating LLMs: HumanEval (164 programming problems) [1], MBPP (500 Python programming problems) [2]
>
> - [1] Chen, Mark, et al. "Evaluating large language models trained on code." arXiv preprint arXiv:2107.03374 (2021).
> - [2] Austin, Jacob, et al. "Program synthesis with large language models." arXiv preprint arXiv:2108.07732 (2021).
>
> Additionally, it's important to highlight several strengths of our dataset:
> - **Uniqueness:** Our dataset fills a gap in our community, as no similar attribution evaluation datasets exist.
> - **Diversity:** Despite its small size, our dataset spans 12 diverse domains.
> - **Timeliness:** It's based on the latest LLMs and generative search engine, New Bing, ensuring timeliness.
> - **Data Contamination:** We created queries from scratch, avoiding the data contamination issue commonly seen with existing LLMs..
> - **Reproducibility:** We released our annotation protocol and interface in the Appendix, enabling future researchers to replicate our process and annotate more datasets.
>
> With that said, we would take expanding the dataset size as one of our future steps, as noted in the paper (Line 551-558).
>
> ---
>
> **Presentation Improvements**: We appreciate your suggestions for improving the presentation of our paper. We will correct the typo you've pointed out and move more data repurposing details from Appendix A to the main content.

---

### Official Review · Reviewer_BA69 · 2023-08-12

**Soundness:** 3

**Excitement:**

3: Ambivalent: It has merits (e.g., it reports state-of-the-art results, the idea is nice), but there are key weaknesses (e.g., it describes incremental work), and it can significantly benefit from another round of revision. However, I won't object to accepting it if my co-reviewers champion it.

**Paper Topic And Main Contributions:**

This article presents two new approaches for the automatic evaluation of LM's ability to provide answers, which are supported by external references. The article presents a new classification of LM attribution, composed of 3 classes, and explains how data is collected.

Contributions:

1. New classification of types of attributions.
2. Two new methods for automatic evaluation of the attribution ability of LMs.

**Reasons To Accept:**

1. New classification of types of attributions.
2. Two new methods for automatic evaluation of the attribution ability of LMs.

**Reasons To Reject:**

The aim of the article is unclear - is it presenting methods to collect a new dataset, or the actual evaluation approaches?

**Reproducibility:**

3: Could reproduce the results with some difficulty. The settings of parameters are underspecified or subjectively determined; the training/evaluation data are not widely available.

**Reviewer Confidence:**

3: Pretty sure, but there's a chance I missed something. Although I have a good feel for this area in general, I did not carefully check the paper's details, e.g., the math, experimental design, or novelty.

---

> ### Author Rebuttal · Authors · 2023-08-29
>
> We would like to express our gratitude for your feedback to our paper. We are pleased that you recognized the contributions of our paper and would like to address the confusion over the aim of our work here:
>
> **Clarification of the Paper's Aim**: Our paper presents the first comprehensive framework for evaluating attribution in LLMs, which includes **three interconnected components:**
>
> - **A new definition of attribution types** (Lines 073-088): We propose a more fine-grained categorization for attribution, identified as attributable, extrapolatory, and contradictory. This nuanced classification not only enhances the safety of LLM usage but also provides critical insights for future mechanisms tailored to correct specific errors.
>
> - **Two approaches for automatic evaluation of attribution** (Lines 089-093): Based on our proposed definition, we explore two methods for identifying and categorizing the attribution errors: prompting LLMs and fine-tuning language models on simulated and repurposed data from related tasks.
>
> - **The introduction of two new datasets for evaluating attribution** (Lines 96-105): To facilitate the evaluation of attribution, we propose two test sets: AttrEval-Simulation and AttrEval-GenSearch. The former is a synthetic dataset repurposed from related tasks, and the latter is an annotated dataset curated from queries in a generative search engine, New Bing. These datasets not only provide a comprehensive testing ground for our evaluation approaches but also contribute to the broader research community by filling the gap in attribution evaluation datasets.

---

### Meta-Review · Area_Chair_pKtL · 2023-09-18

**Recommendation:** 4

**Metareview:**

The paper focuses on using LLMs for automatic evaluation of attribution. Specifically, it makes three contributions: (a) a new NLI-inspired categorization of attribution types, (b) new datasets (simulated and real) for evaluating attribution, and (c) models for automatic evaluation of attribution.

This paper addresses an important problem and provides reasonable yet clearly imperfect solutions. Specifically, reviewers have concerns that their simulated data are biased and that their "real" test dataset is too small. NLP has had a fairly established precedent of using simulated data (especially for training) in settings where human data collection is costly. Typically, such data are indeed biased in certain ways, but are quite often still useful at least until a large costly human-curated dataset becomes available. The concern that their real/unsimulated evaluation set is small (~250 examples) is a more serious weakness.  Nonetheless, I think a fair bit of work has gone into the paper, and it will likely generate more followup work. Overall, I'd be inclined to see it accepted. I also hope that the authors will consider expanding the size of the evaluation set to make this paper stronger.

---

### Decision · Program_Chairs · 2023-10-07

**Decision:**

Accept-Findings

**Comment:**

The paper focuses on using LLMs for automatic evaluation of attribution. Specifically, it makes three contributions: (a) a new NLI-inspired categorization of attribution types, (b) new datasets (simulated and real) for evaluating attribution, and (c) models for automatic evaluation of attribution.

This paper addresses an important problem and provides reasonable yet clearly imperfect solutions. Specifically, reviewers have concerns that their simulated data are biased and that their "real" test dataset is too small. NLP has had a fairly established precedent of using simulated data (especially for training) in settings where human data collection is costly. Typically, such data are indeed biased in certain ways, but are quite often still useful at least until a large costly human-curated dataset becomes available. The concern that their real/unsimulated evaluation set is small (~250 examples) is a more serious weakness.  Nonetheless, I think a fair bit of work has gone into the paper, and it will likely generate more followup work. Overall, I'd be inclined to see it accepted. I also hope that the authors will consider expanding the size of the evaluation set to make this paper stronger.